# An Innovative Intervention Model for Children and Young People in Residential Care: The D’ART-TE Project

**DOI:** 10.3390/healthcare12222237

**Published:** 2024-11-10

**Authors:** Fátima Gameiro, Miguel Faria, Beatriz Rosa, Paula Ferreira, Ana Pedro

**Affiliations:** 1LusoGlobe, Institute of Social Work, Lusófona University—University Center of Lisbon, Campo Grande, 376, 1749-024 Lisbon, Portugal; paula.ferreira@ulusofona.pt; 2Ribeiro Sanches School of Health, Lusophone Polytechnic Institute, 2500-447 Lisbon, Portugal; miguel.faria@ipluso.pt; 3School of Psychology and Life Sciences, Lusófona University—University Center of Lisbon, Campo Grande, 376, 1749-024 Lisbon, Portugal; beatriz.rosa@ulusofona.pt; 4Santa Casa da Misericórdia de Santarém, 2000-135 Santarém, Portugal; ana.pedro@scms.pt

**Keywords:** residential care, children and youth, intervention model, D’AR-TE

## Abstract

(1) Background: Residential care (RC) for children and youth should provide a holistic experience of growing up. Currently, RC homes face many challenges, highlighting the need for validated, innovative interventions. D’AR-TE is a pilot project aimed at creating an innovative intervention model for children and youth in RC, promoting protective factors through activities designed to enhance personal skills, socialization, and relationship building. The project spanned three years, focusing on two main areas: “Promoting the SELF” (including Sports, Art, and Virtual Reality) and “Promoting the US” (group activities with families and non-institutionalized peers). (2) Methods: To validate the D’AR-TE model, 46 children and youth in RC, aged between 6 and 20 years, were assessed. They were divided into an experimental group (26 participants) and a control group (20 participants). The evaluation, conducted at the beginning and end of the project, focused on cognitive and neuropsychological domains. (3) Results: By the end of D’AR-TE, the experimental group showed statistically significant improvements, including decreased aggression perception, increased social support, enhanced self-concept, and better cognitive and emotional performance. (4) Conclusions: These results suggest that D’AR-TE had a positive impact and could be a valid and replicable model for children and youth in RC.

## 1. Introduction

According to the United Nations Children’s Fund (UNICEF) [1], in 2024, 456,000 children live in institutions. Portugal follows the trend observed in Central Europe, which has the highest rate of children in residential care (RC) units, 294/100,000 children. In Europe and Central Asia, Portugal has the highest number of institutionalized children, presenting almost three times the world average. According to the most recent CASA report (ISS) [2], there were 6347 children and youth in care in Portugal, predominantly male (52%). In RC, behavioral problems are the most prevalent (25%), with a higher incidence among males (60.8%), and mental health-related issues (14%). About 38% received psychological support, 27% had child psychiatry/psychiatry care, and approximately 28% were on psychiatric medication.

Violence and aggression have been increasingly prominent among these children and youth. They are often exposed to dysfunctional and inadequate physical and emotional care and/or deprived of secure relationships with peers and adults. Prior experiences before RC often involve inadequate healthcare and hygiene, limited expression of affection, and fragile interpersonal relationships within disorganized family dynamics that increase the likelihood of aggressive behaviors as a way to cope with negative emotions such as fear, anguish, anger, and hatred. Moreover, placement in a non-family context and prolonged separation from their family can act as additional factors of emotional destabilization [3,4,5,6,7,8]. Also, Montserrat et al. carried out a study in 2021 [9] with 33 early adolescents in RC in Portugal and Spain and concluded that the young people showed dissatisfaction concerning matters of individuality, autonomy, participation, and socialization.

Behavioral problems, specifically peer aggression in RC, can assume considerable significance. According to Silva [10], these behaviors may be increased by exposure to perpetrators, as they share the same home, allowing easy physical access and personal information. These conditions can facilitate the aggressors’ tendency to harm, humiliate, or control, aiming to obtain rewards such as a sense of control and power, elevation of status among peers, emotional relief, entertainment, or material gains. These experiences demand constant adaptability, with a likelihood of aggressive behaviors manifesting as the child or youth struggles to find their place in the new environment. The literature indicates that children and youth with at least one official record of maltreatment between birth and age 10 have a significantly higher frequency of aggressive behaviors during adolescence [11].

Psychiatry and psychotherapy have been the most widely used to mitigate negative feelings and control aggressive behaviors in this population [12,13,14,15]. In 2022, Cameron-Mathiassen’s team [16] conducted a systematic literature review with the aim of identifying the experience of living in RC. They searched five relevant databases for qualitative empirical studies published between 1990 and January 2020. The results were heterogeneous. For some, RC is referred to as an experience that promotes care, stability, security, and a perception of support, while others report experiences of not being listened to or understood by the institution and a poor perception of well-being. Relationships with peers were also experienced both positively, with the development of friendly and supportive relationships, and negatively, with the peer group accepting bullying and violence as normative behaviors. According to Henderson [17] several authors state that RC as a treatment of choice with certain groups of needy children and youth, not a last resort intervention. Therefore, it is crucial to consider, propose, and test new forms of intervention that may prove more effective than those currently in use. In this line, the D’AR-TE project aimed to be an alternative to this type of intervention.

### D’AR-TE Project: Presentation and Characterization

For three years, the D’AR-TE project aimed to promote salutogenic behavior, enhance resilience, and indirectly reduce aggressive behaviors. The organizing entity (Santa Casa da Misericórdia de Santarém/SCMS) proposed an innovative and differentiated initiative, offering a set of diverse activities (Virtual Reality, Judo, Arts) that facilitate protective factors. The promotion of these factors, in the medium to long term, can prevent and minimize risk behaviors and instances of aggression, thereby contributing to the development of personal skills, interpersonal socialization, and the strengthening of relationships among peers, family, and community [18].

D’AR-TE is an innovative project in terms of its intervention model with children and youth in RC. To our knowledge, no other project, either national or international, offers these particular components. The project ran from March 2020 to March 2023, with the partnership of several local institutions, which acted as social investors.

The activities of the D’AR-TE project were organized around two main axes: “Promoting the SELF” and “Promoting the US.” The first axis included Sports (judo), an Arts space (theater/short film, music, and body expression/dance), and Virtual Reality (development, and use of innovative virtual reality tools to promote cognitive functions). The second axis integrated group activities involving families and non-institutionalized peers.

The choice of judo was justified by literature that highlights its potential to enhance physical, behavioral, cognitive, and socio-emotional skills. Judo allows children and youth to channel their energy positively, both internally and externally, within playful and interactive contexts [19,20]. The practice of judo aimed to improve motor coordination, body control, teamwork, healthy competition, self-confidence, self-concept, self-esteem, problem-solving, inhibitory control, and planning. It also sought to promote respect for others and the practice environment, fostering a culture of self-mastery, reducing impulsive behaviors and violence, and enhancing conscious decision-making [21].

The arts, specifically theater/acting, music, and body expression, were also included because they contribute to the personal and social identity construction of children and youth. These activities enhance a better understanding of cultural issues and promote psycho-emotional balance (self-esteem, positive self-concept, and perceived self-efficacy) and relational well-being [22]. The program involved workshops in theater/films, music, and body expression/dance, fostering assertive communication strategies, teamwork, and promoting self-confidence, self-concept, self-esteem, and creativity stimulation. It aimed to provide a space for the development of expressiveness among children and youth in personal, social, and cultural spheres, serving as an educational, inclusive, and therapeutic component [23,24,25].

The inclusion of virtual reality (VR) was supported by Luria’s Functional Systems Theory, particularly the concept of brain plasticity [26]. The aim was to reproduce various aspects of daily life through an immersive, systematic process, oriented towards activities that can follow distinct approaches and primarily aim to improve cognitive functioning and indirectly foster balanced holistic development [27]. The VR scenarios were specifically developed for the D’AR-TE project in partnership with Hei-Lab. These scenarios included tasks with progressively increasing difficulty. The goal was to innovate by cognitively stimulating children and youth through an immersive environment, aligning with their interests and motivations while simultaneously enhancing their cognitive abilities, which literature identifies as often compromised and influencing their (im)balance [28]. Specifically, the project aimed to improve performance in cognitive functions such as attention, memory, and executive functions (inhibitory control, cognitive flexibility, decision-making/problem-solving and planning).

The aim of the group activities with families was to enhance the effectiveness of sharing, promote assertive communication between parents and children, as well as among families, and to strengthen the establishment of educational boundaries between adults and children/youth. The intervention with non-institutionalized peers, by integrating them into some of the D’AR-TE activities and in outdoor bootcamps, aimed to break cycles of relational and social isolation and reduce the social stigma of institutionalization.

The objective of this work was to validate the D’AR-TE project as an intervention model for children and youth in RC, based on the practice of judo, art, and VR, along with group activities involving family and peers. The goal was to promote personal skills, interpersonal socialization, and the stimulation of relationships.

## 2. Materials and Methods

### 2.1. Participants

A sample of 46 children and youth in RC participated in this study, with ages ranging from 6 to 20 years old (M = 13.9, SD = 4.1), the majority of which were male (96.2%). Of these, 26 (56.5%) were assigned to the experimental group and 20 (43.5%) to the control group (see Table 1). The experimental group was constituted by children and young people with a promotion and protection measure who were in RC at the institution where D’AR-TE was implemented. To constitute the control group, a survey was carried out of RC homes with a population as similar as possible to that of the experimental group (mostly males, average of age, reasons for signaling danger).

### 2.2. Instruments

Socio-demographic data were collected through a questionnaire with data about age, gender, institution, and time of integration in this project.

The assessment included a large number of tests on two levels. At the within-subjects level, at the beginning and end of the project (3 years later) and at the between-groups level, four RC units were compared (two as part of D’AR-TE and two others as control group). All the tests used show adequate psychometric qualities and are adjusted to the age of our sample. Given the large number of instruments and variables (55), we have chosen to present them in a table (see Table 2).

The assessment was organized according to the ages of the children and youth and evaluated in the cognitive and neuropsychological domains (cognitive and emotional functions). In each domain, the respective analysis variables were identified, as shown in Table 2.

### 2.3. Procedure

After the approval of study procedures by the ethical committee of ISW Department of Lusófona University (ULCUL-05/1721), participants were recruited from four institutions of central districts of Portugal. The participants from Lar dos Rapazes and Primeiro Passo were assigned to the experimental group, after informed consent has been obtained from legal guardians. The participants from Centro de Bem-Estar Social da Zona Alta/CBESZA and Associação de Melhoramentos e Bem-Estar Social de Pias/Frazoeira were assigned to the control group, having no intervention. To avoid potential biases in the assessment, participants of the control group were unaware of the D’AR-TE promotion. There was no experimental mortality in any of the groups. After initial assessment, both groups were compared to check if they were similar, and being the case, both groups were assessed after the intervention, this time using a pairwise means comparison, to see what evolution had occurred from the first moment (before the intervention) to the second moment (after the intervention).

The initial and final assessments were carried out in two sessions in order to control the effect of fatigue. The application and scoring tests were carried out by experienced clinical psychologists and neuropsychologists and was promoted during the school term and in the middle of the week in order to minimize the possible effects of trips to/from home and vacations on the results.

Due to the COVID-19 pandemic, the project’s schedule had to be readjusted.

The implementation of the D’AR-TE project, which was part of the children’s and young people’s daily routines and took place on RC premises, is shown in Table 3.

### 2.4. Data Analysis

Statistical analysis was performed using the Statistical Package for the Social Sciences (IBM, SPSS Statistics, version 28.0 of Windows). 

At the beginning of the intervention, both groups were compared across the several scales used in our assessment protocol, to check if they could be considered equivalent. The methodology used was an independent mean group’s comparison. Of the 55 variables analyzed, only 9 showed statistically significant differences between the groups, as reported in Table 4. Differences between the control and experimental groups were rare.

After the intervention, both groups were reassessed, using a pairwise mean comparison, where each subject provided two values, one before and one after the intervention. The statistical significative results are presented in Table 5, Table 6 and Table 7.

## 3. Results

### 3.1. Perception of Aggression 

The children’s perceived assertiveness was assessed by the CABS. No statistically significant differences were found within-subjects or between-groups (see Table 5).

As for the youth’s perception of aggression, assessed by the AQ, statistically significant differences were found within-subjects in the experimental group, which showed less physical and verbal aggression (t(18) = 2.504; *p* = 0.022) at the end of the project (M = 2.79; SD = 0.79) compared to the beginning (M = 3.12; SD = 0.69) and less total aggression (t(18) = 2.21; *p* = 0.040) at the end of the project (M = 2.77; SD = 0.62) compared to the beginning (M = 3.00; SD = 0.59). In the control group, no statistically significant differences were found within-subjects (see Table 5).

While no improvements were found in the control group, the results show that the D’AR-TE project seems to reduce aggression among young people, specifically physical and verbal aggression.

### 3.2. Perception of Social Support

In terms of children’s perception of social support (assessed by ‘Era Uma Vez’), no statistically significant differences were found within-subjects or between-groups in any of the variables (see Table 5).

For young people’s perception of social support (assessed by the PSS/Am and PSS/Fam), statistically significant differences were found between within-subjects in the experimental group. They had a higher perception of social support from friends (t(20) = −2.771; *p* = 0.012) at the end of the project (M = 15.19; SD = 2.40) compared to the beginning (M = 13.43; SD = 2.69) and a higher perception of social support from family (t(20) = −2.703; *p* = 0.014) at the end of the project (M = 15.24; SD = 3.74) compared to the beginning (M = 13.76; SD = 4.42). In the control group, no statistically significant differences were found within-subjects (see Table 5).

The young people who participated in D’AR-TE had a higher perception of social support from friends and family. Given that there were no changes in the control group, the results show that the D’AR-TE seems to increase the perception of social support from friends and family.

### 3.3. Self-Esteem

The children’s perception of self-esteem was assessed by ‘Era Uma Vez’ and statistically significant differences were only found in the experimental group between the start and end of the project in one of the assessment items (5.1), with the children showing a higher perception of self-esteem (t(9) = 2.449; *p* = 0.037) at the start of the project (M = 0.70; SD = 0.67) compared to the end (M = 0.30; SD = 0.67) (see Table 5). 

As for young people (assessed by RSS), no statistically significant differences were found in within-subjects or between-groups comparation (see Table 5).

Contrary to what we might have expected, at the end of D’AR-TE the children had a lower perception of self-esteem than at the beginning of their participation, which leads us to conclude that the D’AR-TE project may not promote this domain. 

### 3.4. Self-Concept

As for the children’s perception of self-concept (assessed by ‘Era uma vez’), no statistically significant differences were found in within-subjects or between-groups comparation (see Table 5).

With regard to the young people assessed using the PHCSCS-2, statistically significant differences were found in the experimental group between the start and end of the project in all the dimensions assessed, in the sense of positive changes at the end of the project compared to the start, particularly in anxiety (t(21) = 2. 916; *p* = 0.008), physical appearance (t(21) = 2.873; *p* = 0.009), behavior (t(21) = −3.554; *p* = 0.002), popularity (t(21) = −3.343; *p* = 0.003), satisfaction/happiness (t(21) = −3.725; *p* = 0.001), intellectual status (t(21) = −3.202; *p* = 0.004), and total self-concept (t(21) = −4.219; *p* < 0.001) (see Table 5). In the control group, statistically significant differences were found in within-subjects in popularity (t17) = −2.719; *p* = 0.015), which was higher at the end of the project (M = 4.44; SD = 1.15) compared to the beginning (M = 3.97; SD = 1.13) and in total self-concept (t(17) = −2.117; *p* = 0.049), which was also higher at the end of the project (M = 4.01; SD = 0.72) compared to the beginning (M = 3.80; SD = 0.61) (see Table 5).

Young D’AR-TE participants showed less anxiety, better physical appearance, more adapted behavior, more popularity, more satisfaction/happiness, higher intellectual status, and higher total self-concept. However, those in the control group also showed more popularity and a better total self-concept. It is important to consider that in the area of total self-concept, the young people showed statistically significant differences in the initial comparison (*p* = 0.020) between the experimental group (M = 4.27; SD = 0.59) and the control group (M = 3.80; SD = 0.61).

While no improvements were found in the control group in terms of anxiety, physical appearance, behavior, satisfaction/happiness, and intellectual status, the results show that the D’AR-TE project seems to increase the perception of self-concept in these domains.

### 3.5. Cognitive Functions

#### 3.5.1. Perceptual Organization

As for perceptual organization, as assessed by the result of copying the FCR, statistically significant differences were found in the experimental group between the start and end of the project (t(21) = −2.86; *p* = 0.009), with the results being higher at the end (M = 27.05; SD = 9.98) compared to the start (M = 24.07; SD = 8.47), which shows a higher quality of perceptual organization. In the control group, no statistically significant differences were found in within-subjects (see Table 6).

Participants of D’AR-TE showed greater perceptual–motor organization at the end of the project. These results show that participation in this project seems to promote perceptual organization, specifically the quality of children and young people’s perceptual analysis.

#### 3.5.2. Verbal Fluency Capacity

The children’s verbal fluency (assessed by TI-BAFEC Verbal Fluency) no statistically significant differences were found in within-subjects or between-groups comparations (see Table 6).

As for the verbal fluency of the young people in the experimental group (assessed by Lexical and Semantic Fluency), no statistically significant differences were found in within-subjects. However, in the control group, statistically significant differences within-subjects were found in lexical fluency (t(16) = −2.297; *p* = 0.035), with the results being higher at the end of the project (M = 7.65; SD = 3.77) compared to the start (M = 6.00; SD = 3.02) (see Table 6). The young people of control group showed greater lexical fluency, demonstrating that the D’AR-TE doesn’t seem to boost children and young people’s verbal fluency.

#### 3.5.3. Attention 

As for the children’s attention (assessed using the BAFEC TI/Simple Attention score), no statistically significant differences were found within-subjects or between-groups (see Table 6).

With regard to young people’s attention span (assessed using CTT-1 and CTT-2, and TP/Dispersion Index and TP/Work Performance), there were only statistically significant differences within-subjects in CTT-1 (t(21) = 2.445; *p* = 0.023) with shorter execution time at the end of the project (M = 59.14; SD = 30.35) compared to the beginning (M = 73.91; SD = 40.11) and in CTT-2 (t(21) = 4.911; *p* < 0.001) with shorter execution time at the end of the project (M = 130.00; SD = 61.31) than at the beginning (M = 167.41; SD = 86.98). No statistically significant differences were found within-subjects in the control group of young people (see Table 6).

The young participants in D’AR-TE showed an improvement in their attentional patterns which allow us to conclude that this project promoted attention in different domains (sustained, visual search, divided and alternation) in the young people, but did not have the same effect on the children’s performance. 

#### 3.5.4. Memory

As for the children’s memory (composite score of the TI-BAFEC/Simple Memory), no statistically significant differences were found within-subjects or between-groups comparations (see Table 6).

The memory of the young people was assessed using the FCR/memory score and a dimension of the WISC-III/Digit Memory in the direct sense. In the experimental group, there were statistically significant differences within-subjects in the FCR/memory (t(21) = −5.364; *p* < 0.001), with the results being higher at the end of the project (M = 15.78; SD = 7. 63) compared to the beginning (M = 8.82; SD = 6.62); and in the WISC-III/Digit Span Memory (t(16) = −5.416; *p* < 0.001) with higher results at the end of the project (M = 11.12; SD = 3.43) compared to the beginning (M = 8.53; SD = 4.05). In the control group of young people, no statistically significant differences were found within-subjects (see Table 6).

The young participants of D’AR-TE increased their visual memory and working memory. These results show that the D’AR-TE promoted memory, specifically visual memory and working memory in young people.

#### 3.5.5. Cognitive Flexibility

Cognitive flexibility (assessed by WCST scores), showed statistically significant differences within-subjects in the experimental group in perseverative responses (t(20) = 4.327; *p* < 0.001), with the lowest scores at the end of the project (M = 8.52; SD = 10.49) compared to the beginning (M = 21.86; SD = 9.72). There were also statistically significant differences within-subjects in the number of categories completed (t(21) = −5.631; *p* < 0.001), with more categories completed at the end (M = 5.73; SD = 0.77) than at the start (M = 3.68; SD = 1.62) of the project. The decrease in perseverative responses and the increase in the number of categories completed are indicative of better cognitive flexibility at the end of the project. In the control group, statistically significant differences within-subjects were also found in perseverative responses (t(17) = 2.325; *p* = 0.033), with the results being lower at the end of the project (M = 17.22; SD = 15.19) compared to the beginning (M = 23.17; SD = 11.11), which shows an improvement (see Table 6).

The young people who participated in D’AR-TE showed more cognitive flexibility. However, the young people in the control group also showed an improvement in cognitive flexibility at the end of the project. As no differences were found in the control group for the number of categories completed, we can conclude that participation in the D’AR-TE project seems to favor the cognitive flexibility of children and young people as they were able to learn and respond successfully to contextual changes. 

#### 3.5.6. Inhibitory Control

In inhibitory control (assessed using the STROOP Test and two FAB tests-SI and CI), there were only statistically significant differences within-subjects in the experimental group in FAB/CI (t(21) = −3. 196; *p* = 0.004), with the results being higher at the end of the project (M = 2.86; SD = 0.35) compared to the beginning (M = 2.32; SD = 0.84), which reveals greater inhibitory control of the D’AR-TE participants. As for the control group, no statistically significant differences within-subjects were found in any of the tests used (see Table 6).

As no improvements were found in the control group, the results show that the D’AR-TE project seems to enhance the inhibitory control capacity of children and young people.

#### 3.5.7. Planning

As for the children’s ability to plan (assessed using the TI-BAFEC/Planning), there were no statistically significant within-subjects or between-groups (see Table 6).

In the young people, planning was assessed by three ToL scores. In the experimental group, there were statistically significant differences within-subjects in the total correct score (t(21) = −3.696; *p* < 0.001), with higher results at the end of the project (M = 3.05; SD = 1.43) compared to the start (M = 1.73; SD = 1.32). There were also statistically significant differences within-subjects in the number of movements (t(21) = 5.079; *p* < 0.001), with lower results at the end of the project (M = 35.09; SD = 16.31) compared to the start (M = 52.86; SD = 17.37). For total execution time, there were also statistically significant differences within-subjects (t(21) = 3.818; *p* = 0.001), with shorter execution time at the end of the project (M = 232.05; SD = 60.79) compared to the start (M = 325.18; SD = 118.39). At the end of D’AR-TE there was a greater number of problems solved correctly, using fewer moves and in less time, indicating a higher planning capacity. In the control group, statistically significant differences were found within-subjects in only one of the ToL results, the total execution time (t(18) = 2.303; *p* =.033), with shorter time at the end of the project (M = 219.89; SD = 56.11) compared to the start (M = 264.84; SD = 90.58), which shows that the participants had improved in terms of how fast they could execute a plan (see Table 6).

The young people who took part in D’AR-TE had better planning skills at the end of the project. The control group also showed better planning skills at the end of the project, but only in one of the three dimensions. These results show that the D’AR-TE project seems to enhance the quality of planning and the speed of execution of young people.

#### 3.5.8. Verbal Intelligence

In terms of verbal intelligence (assessed using three WISCIII scales), there were statistically significant differences within-subjects in the comprehension test in the experimental group (t(18) = −2.528; *p* = 0.021), with the results being higher at the end of the project (M = 5.00; SD = 3.11) compared to the beginning (M = 3.05; SD = 3.22), which shows a greater capacity for verbal intelligence on the part of the participants in D´AR-TE. In the control group, statistically significant differences were found within-subjects in the comprehension test (t(17) = −3.828; *p* = 0.001), with the results being higher at the end of the project (M = 5.11; SD = 3.41) compared to the beginning (M = 2.61; SD = 2.81), which also shows an improvement (see Table 6).

Participants of D’AR-TE had higher verbal intelligence scores. However, the control group also showed higher results at the end of the project, which shows that the D’AR-TE project does not seem to boost this dimension.

### 3.6. Emotional Functions

#### 3.6.1. Empathy

As for empathy (assessed using the composite score of the TI-BAFEC/Theory of Mind), there were no statistically significant differences within-subjects or between-groups, so they did not show any improvement in this dimension.

The results show that the D’AR-TE project does not seem to increase empathy/ability to understand different points of view in children and young people.

#### 3.6.2. Understanding Irony 

Regarding the comprehension of irony (assessed using the TI-BAFEC/comprehension of irony), the experimental group found statistically significant differences within-subjects (t(25) = −2.205; *p* = 0.037), with the results being higher at the end of the project (M = 8.15; SD = 3.30) compared to the beginning (M = 7.14; SD = 2.78), which shows a greater ability to understand irony at the end of the project. As for the control group, there were no statistically significant differences within-subjects (see Table 7).

The results show that the D’AR-TE project seems to promote children and young people’s ability to understand the non-literal meaning of language.

#### 3.6.3. Emotional Decision

For emotional decision-making ability (assessed by TI-BAFEC/emotional decision-making), the experimental group found statistically significant differences within-subjects in the emotional decision-making test (t(25) = 3.423; *p* = 0.002), with higher results at the start of the project (M = 75.22; SD = 12.58) compared to the end (M = 67.01; SD = 7.36), which shows greater emotional decision-making ability at the end of the project. As for the control group, there were no statistically significant differences within-subjects (see Table 7).

D’AR-TE participants showed a greater capacity for emotional decision-making (TI-BAFEC/Emotional decision-making; *p* = 0.002). It is important to consider that in the area of emotional decision-making, the children/young people showed statistically significant differences in the initial comparison (*p* = 0.016) between the experimental group (M = 75.22; SD = 12.58) and the control group (M = 67.07; SD = 8.23). As there were no changes in the control group, the results show that the D’AR-TE project seems to strengthen and increase the emotional decision-making capacity of children and young people. This result reflects the ability to make advantageous decisions quickly in a situation where a rational analysis of all the factors involved in a given situation is not possible.

## 4. Discussion

According to Anglin [46], the purpose of a RC is to create an artificial living environment that offers the youth residents an opportunity to develop a sense of normality. In Portugal, the current model for intervention in RC is referred to as therapeutic, however, the results regarding risk and danger behaviors in this population have been increasingly worrying and unlike what happened in the past (which stemmed from the behaviors of caregivers, e.g., neglect, domestic violence), the danger comes mostly from behaviors adopted by the children and young people themselves. The D´AR-TE project, based on promoting protective factors (proactive/preventive rather than therapeutic), is innovative and fundamental.

Based on the analysis of the results, even knowing that there are other variables that may have influenced the results, we can argue that the objective of D’AR-TE has been achieved. The perceived decrease in aggression among young people, specifically physical and verbal aggression, an increase in the perception of social support (greater perception of support from the young people’s family and friends), self-concept (anxiety, physical appearance, behavior, satisfaction/happiness, and intellectual status), and an increase in cognitive performance in general, both in terms of cognitive functions (perceptual organization; attention; memory; cognitive flexibility; inhibitory control and planning), as well as emotional functions (ability to understand irony and emotional decision-making) were more evident in the young people who took part in D’AR-TE project.

The effect of the implementation of D’AR-TE, specifically the promotion of arts workshops (theater, music and body expression), sports (judo), cognitive stimulation (virtual reality) and mobilization/intervention with families and peers, in an integrated and systemic way, was positive for the children and young people in RC because it allowed them to reduce aggressive behavior and promote emotional and cognitive skills. The new RC intervention model proved to be effective in these children and young people, however, the extrapolating these findings to other settings should be done with caution. The follow-up evaluation (one year after the end of the project) is still running.

In the implementation of D’AR-TE there are some aspects that should be mentioned and pointed out as limitations of the study. Firstly, the sample size and type, with only four RC Units and a total of 46 children and youth, does not allow us to extrapolate to all the children and young people sheltered in RC units. The fact that two of them only take in boys and the other two are mixed created a sample that was not very significant or representative in terms of gender, although as we mentioned earlier, the majority of children and young people in care in Portugal are male.

The use of different tests to assess the same domain can be also criticized, but we opted for this alternative because the participants in the D’AR-TE had a wide age range. To control this variable, all the tests included in the assessment protocol guaranteed adequate psychometric qualities and were validated for the respective age groups, and precautions were taken to comply with the ethical and conduct issues adopted by psychology in assessment processes.

Although questionnaires are a common way of getting to know children and young people in RC, it is possible that the participants’ responses were influenced by factors such as self-image or the desire to provide socially desirable answers. In future studies we suggest including the assessment of the domains considered in the D’AR-TE by hetero-assessment (e.g., caregivers, technicians, family members) and using concomitant qualitative measures in order to increase the robustness of the results.

Future studies may consider replicating this intervention model in other RC units to support these results and consolidate its robustness. It would also be important to promote this project in RC settings that include more girls to validate its effectiveness with this population.

## Figures and Tables

**Table 1 healthcare-12-02237-t001:** Sociodemographic characteristics of the sample (N = 46).

	Experimental Group		Control Group		Total	
Age	n	%	n	%	n	%
6–7 years	4	15.4	1	5.0	5	10.9
8–10 years	4	15.4	2	10.0	6	13.0
11–14 years	7	26.9	4	20.0	11	23.9
15–17 years	5	19.2	10	50.0	15	32.6
18–20 years	6	23.1	3	15.0	9	19.6
Gender	n	%	n	%	n	%
Male	25	96.2	20	100	45	97.8
Female	1	3.8	0		1	2.2
Total	26	100.0	20	100.0	46	100.0

**Table 2 healthcare-12-02237-t002:** D’AR-TE: assessment protocol by cognitive and neuropsychological domain according to age and respective variables.

Domain	Evaluation	Instruments—Age Group	Variables
Cognitive	Perception of Aggression	6–10 years: Children’s Assertive Behavior Scale (CABS) (Michelson & Wood, 1982) [29]	Total score; Passive score; Aggressive score
		11–20 years: Aggression Questionnaire (AQ) (Buss & Perry, 1992) [30]	Total score; Instrumental; Affective; Cognitive
	Perception of Social Support	6–10 years: Children’s Social Support from Parental Figures and Peers (Semi-projective test-‘Era Uma Vez’ (Fagulha, 1992) [31]	Parental figures (Items 1.1.; 1.2.; 1.3.) Peers (Items 3.1.; 3.2.; 3.3)
		11–20 years: Perception of Social Support from Family (PSS-Fam) and from Friends (PSS-Par) (Procidano & Heller, 1983) [32]	PSS Fam; PSS_Par
	Self-Esteem	6–10 years: Children’s Self-Esteem (‘Era Uma Vez’) (Fagulha, 1992) [31]	Self-esteem (Items 5.1.; 5.2.; 5.3.)
		11–20 years: Rosenberg Self-Esteem Scale (RSS) (Rosenberg, 1965) [33]	RSS
	Self-Concept	6–10 years: Children’s Self-Concept (‘Era Uma Vez’) (Fagulha, 1992) [31]	Self-concept (Items 7.1.; 7.2.; 7.3.)
		11–20 years: Piers–Harris Children’s Self-Concept Scale (PHCSCS-2) (Piers & Herzberg, 2002) [34]	Anxiety; Physical appearance; Behavior; Popularity; Satisfaction/Happiness; Intellectual status; Total self-concept
Neuropsychological	Neuropsychological Assessment Protocol (NA)	6–7 years: ‘Tartaruga da Ilha’-Battery for the Assessment of Executive Functions in Children (TI-BAFEC) (Mesquita, 2011) [35]	Cognitive Functions: Items 1, 2, 4, 9 and 19
			Emotional Functions: Items 14, 17, 18 and 20
		8–10 years: TI-BAFEC (Mesquita, 2011); The Color Trail Test (CTT) (D’ Elia et al., 1996) [36]; Wisconsin Card Sorting Test (WCST) (Kongs et al., 2000) [37]; Stroop Color and Word Test (Stroop) (Stroop, 1935) [38]; Frontal Assessment Battery—Interference Sensitivity and Inhibitory Control (Dubois et al., 2000) [39]; Tower of London (ToL) (Shallice, 1982) [40]; Wechsler Intelligence Scale for Children-5 tests (WISC-III) (Wechsler, 2003) [41]; Rey–Osterrieth Complex Figure Test (FCR) (Rey, 1941; Osterrieth, 1944) [42,43]	Cognitive Functions: Verbal fluency: TI-BAFEC (Items 1 and 2) Perceptual-motor organization: FCR (Copy score) Attention: TI-BAFEC (Item 4); CTT-1 (Sustained Attention Time); CTT-2 (Divided Attention Time) Memory: FCR (Memory score); WISC-III (Digits memory) Cognitive flexibility: WCST (Number of answers p; Number of categories completed) Inhibitory Control: Stroop (Total); FAB (SI and CI) Planning: TI-BAFEC (item 19); ToL (Total score; Number of movements; Total execution time) Verbal Intelligence: WISC-III (Information; Vocabulary; Comprehension)
			Emotional Functions: TI-BAFEC (Items 14, 17, 18 and 20)
		11–20 years: CTT; WCST; Stroop; FAB-IS; FAB-CI; ToL; FCR; Verbal Fluency Test (VF) (Newcombe, 1969) [44]; Toulouse and Piéron Cancelation Test (TP) (Toulouse & Piéron (1904) [45]; WISC-III; TI-BAFEC/emotional functions.	Cognitive Functions: Perceptive Organization: FCR (Copy score) Verbal fluency: VF (Phonological and Semantic: number of correct words) Attention: TI-BAFEC (Item 4); CTT-1 (Time); CTT-2 (Time); TP (Dispersion Index; Labor Income) Memory: FCR (Memory score); WISC-III (Digits Memory) Cognitive flexibility: WCST (Number of answers p; Number of categories completed) Inhibitory Control: Stroop (Total); FAB (SI and CI) Planning: TI-BAFEC (item 19); ToL (Total score; Number of movements; Total execution time) Verbal Intelligence: WISC-III (Information; Vocabulary; Comprehension)
			Emotional Functions: TI-BAFEC (Items 14, 17, 18 and 20)

**Table 3 healthcare-12-02237-t003:** Representation of the D’AR-TE implementation.

Experimental Group			Control Group
1st Evaluation (20 March–20 April)			1st Evaluation (20 May–20 June)
Axis “Promoting the SELF”			
Judo (In group)	Art Atelier (In group)	Virtual Reality (Individual)	
Facilitator: Judo Master	Facilitator: Professors	Facilitator: Neuropsychologist	
Frequency: 2 times/week 1 h (6–10 years) 2 h (11–20 years)	Frequency: 1 time/week	Frequency: 30 min/day	
Activities/Duration: Judo sessions (20 September–23 March)	Activities/Duration: Theater/Performance (20 September–21 July) Music (21 September–22 July) Body Expression (23 September–23 March)	Activities/Duration: Cognitive Stimulation (20 September–23 March)	
Axis “Promoting the Us”			
Family	Peers		
Duration: judo, acting, music and body expression activities. Positive parenting sessions (21 October–23 March)	Activities/Duration: Involvement in judo, acting, music and body expression activities Bootcamps (21 October–23 March)		
Frequency: once a month			
2nd Evaluation (23 April)			2nd Evaluation (23 June)

**Table 4 healthcare-12-02237-t004:** Comparison of scale scores before the intervention in both groups.

	Group						
	Experimental		Control				
	Mean	SD	Mean	SD	T	df	*p*
Era-Uma-Vez-Item 1.2.	0.20	0.42	1.33	1.15	−2.764	11	0.018
PHCSCS-2-Total self-concept	4.27	0.59	3.80	0.61	2.432	38	0.020
TI-Item 14	10.08	3.17	11.80	0.62	−2.388	44	0.021
TI-Item 20	75.22	12.58	67.07	8.23	2.509	44	0.016
Color Trails time-2	167.41	86.98	112.05	51.61	2.427	39	0.020
FCR-Copy score	8.82	6.62	13.29	6.93	−2.110	39	0.041
ToL-Total score correct	1.73	1.32	3.11	1.88	−2.746	39	0.009
ToL-Movements	52.86	17.37	37.32	13.09	3.195	39	0.003
WISC III-Information	3.36	2.99	5.83	4.16	−2.183	38	0.035

**Table 5 healthcare-12-02237-t005:** Statistically significant results for the perception of aggression, perception of social support, self-esteem, and self-concept (before and after the intervention).

		**Start**		**End**		**Difference**			
**Experimental Group**	**n**	**Mean**	**SD**	**Mean**	**SD**	**Mean**	**SD**	**t**	** *p* **
AQ-Total score	19	3.00	0.59	2.77	0.62	0.23	0.46	2.21	0.040
AQ-Instrumental	19	3.12	0.69	2.79	0.79	0.34	0.58	2.504	0.022
PSS-Par	21	13.43	2.69	15.19	2.40	−1.76	2.91	−2.771	0.012
PSS-Fam	21	13.76	4.42	15.24	3.74	−1.48	2.50	−2.703	0.014
Era Uma Vez-5.1-Self-esteem	10	0.70	0.67	0.30	0.67	0.40	0.52	2.449	0.037
PHCSCS-2-Anxiety	22	4.57	1.01	3.93	1.19	0.65	1.04	2.916	0.008
PHCSCS-2-Physical appearance	22	4.23	1.28	4.86	1.23	−0.63	1.03	−2.873	0.009
PHCSCS-2-Behavior	22	3.96	1.21	4.73	0.94	−0.76	1.01	−3.554	0.002
PHCSCS-2-Popularity	22	4.28	0.93	4.96	0.89	−0.68	0.95	−3.343	0.003
PHCSCS-2-Satisfaction/Happiness	22	4.07	1.06	4.78	0.89	−0.70	0.89	−3.725	0.001
PHCSCS-2-Intellectual status	22	4.00	1.12	4.77	1.02	−0.77	1.12	−3.202	0.004
PHCSCS-2-Total self-concept	22	4.27	0.59	4.67	0.59	−0.40	0.45	−4.219	<0.001
		**Start**		**End**		**Difference**			
**Control Group**	**n**	**Mean**	**SD**	**Mean**	**SD**	**Mean**	**SD**	**t**	** *p* **
PHCSCS-2-Popularity	18	3.97	1.13	4.44	1.15	−0.48	0.75	−2.719	0.015
PHCSCS-2-Total self-concept	18	3.80	0.61	4.01	0.72	−0.21	0.42	−2.117	0.049

Note: AQ—Buss and Perry Aggression Questionnaire; PSS-Fam—Perception of Social Support from Family; PSS-Par—Perception of Social Support from Friends; ‘Era Uma Vez’—Children’s Self-Esteem; PHCSCS-2—Piers–Harris Children’s Self-Concept Scale.

**Table 6 healthcare-12-02237-t006:** Statistically significant results for cognitive functions—perceptive organization; verbal fluency; attention; memory; cognitive flexibility; inhibitory control; planning and verbal intelligence (before and after the intervention).

		**Start**		**End**		**Difference**			
**Experimental Group**	**n**	**Mean**	**SD**	**Mean**	**SD**	**Mean**	**SD**	**t**	** *p* **
FCR-Copy score	22	24.07	8.47	27.05	9.98	−2.98	4.88	−2.86	0.009
VF-Number of correct words	17	6.00	3.02	7.65	3.77	−1.65	2.96	−2.297	0.035
CTT-1-Time/ Sustained Attention	22	73.91	40.11	59.14	30.35	14.77	28.34	2.445	0.023
CTT-2-Time/ Divided Attention	22	167.41	86.98	130.00	61.31	37.41	35.73	4.911	<0.001
FCR-Memory score	22	8.82	6.62	15.78	7.63	−6.97	6.09	−5.364	<0.001
WISC-III-Digits memory (direct direction)	17	8.53	4.05	11.12	3.43	−2.59	1.97	−5.416	<0.001
WCST-Number of answers p	21	21.86	9.72	8.52	10.49	13.33	14.12	4.327	<0.001
WCST-Number of categories completed	22	3.68	1.62	5.73	0.77	−2.05	1.70	−5.631	<0.001
FAB-Inhibitory Control	22	2.32	0.84	2.86	0.35	−0.55	0.80	−3.196	0.004
ToL-Total score	22	1.73	1.32	3.05	1.43	−1.32	1.67	−3.696	0.001
ToL-Number of movements	22	52.86	17.37	35.09	16.31	17.77	16.41	5.079	<0.001
ToL-Total execution time	22	325.18	118.39	232.05	60.79	93.14	114.43	3.818	0.001
WISC-III-Comprehension	19	3.05	3.22	5.00	3.11	−1.95	3.36	−2.528	0.021
		**Start**		**End**		**Difference**			
**Control Group**	**n**	**Mean**	**SD**	**Mean**	**SD**	**Mean**	**SD**	**t**	** *p* **
WCST-Number of answers p	18	23.17	11.11	17.22	15.19	5.94	10.85	2.325	0.033
ToL-Total execution time	19	264.84	90.58	219.89	56.11	44.95	85.09	2.303	0.033
WISC-III-Comprehension	18	2.61	2.81	5.11	3.41	−2.50	2.77	−3.828	0.001

Note: FCR—Rey–Osterrieth Complex Figure Test; VF—Verbal Fluency Test; CTT—The Color Trail Test; WISC-III—Wechsler Intelligence Scale for Children; WCST—Wisconsin Card Sorting Test; FAB-Frontal Assessment Battery; ToL—Tower of London.

**Table 7 healthcare-12-02237-t007:** Statistically significant results for emotional functions—understanding irony and emotional decisions (before and after the intervention).

		Start		End		Difference			
Experimental Group	n	Mean	SD	Mean	SD	Mean	SD	t	*p*
TI-BAFEC (items 17 and 18)-Understanding Irony	26	7.14	2.78	8.15	3.30	−1.01	2.34	−2.205	0.037
TI-BAFEC (item 20)- Emotional Decision Making	26	75.22	12.58	67.01	7.36	8.20	12.22	3.423	0.002

Note: TI—BAFEC-‘Tartaruga da Ilha’-Battery for the Assessment of Executive Functions in Children.

## Data Availability

Data may be made available upon request.

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
