# Peer review of "An Innovative Intervention Model for Children and Young People in Residential Care: The D’ART-TE Project"

_healthcare, 2024, doi:10.3390/healthcare12222237_

Round 1
Reviewer 1 Report
Comments and Suggestions for Authors
A very well written and coherent piece of work. I have a few questions that I believe may enhance this article further.
1. Page 2, paragraph 3, line 1 I would like to see an explanation of why the approach of residential care in Portugal is to use the medical model eg. medication and psychotherapy (problematising the individual).
2. Materials and methods end of page 3, what was the selection process.
3. Page 4, why were so many instruments selected?
4. Procedure - what was the rationale for the split between the universities.
5. P14 Discussion, 1st paragraph - statement re danger doesn't support the insight academics have into the behaviour of children within residential care. For instance - see the work of James Anglin Pain Based Behaviours.
Author Response
We'd like to start by thanking you for your comments. They were extremely important for improving our paper.
Below, we'll respond to the comments you sent us.
- Page 2, paragraph 3, line 1 I would like to see an explanation of why the approach of residential care in Portugal is to use the medical model eg. medication and psychotherapy (problematising the individual).
Response 1: That's an excellent question. Unfortunately, this is the reality in our country. When RC technicians are faced with children and young people who show negative feelings and aggressive behaviors, they usually follow this procedure, i.e. referral to child psychiatry or psychiatry, which often leads to medication. According to the CASA report, by the Social Security Institute (Portuguese Government), in 2023 “an increase (25%) of children and young people characterized with mental disorder is identified ... Given these problems and the needs identified in terms of mental health, almost 40% of children and young people benefit from regular psychological support... The use of medication and regular child psychiatric or psychiatric care (29% and 28%, respectively) are also significant. If we add regular mental health care to irregular care, the record of psychology and psychiatry care becomes even more significant.” (https://www.seg-social.pt/documents/10152/13326/Relatório_CASA_2023/da8913ce-97e0-4b5d-ae10-bf16c7a88901)
However, to make our manuscript clearer, we have inserted the following sentence into the article on line 64 “Psychiatry and psychotherapy have been the most widely used…”
- Materials and methods end of page 3, what was the selection process.
Response 2: Indeed, the sample is a convenience sample.
To answer the question, the following paragraph was introduced - lines 144-148: “The experimental group was constituted by children and young people with a promotion and protection measure who were in RC at the institution where D'AR-TE was implemented. To constitute the control group, a survey was carried out of RC homes with a population as similar as possible to that of the experimental group (mostly males, average of age, reasons for signaling danger).”
- Page 4, why were so many instruments selected?
Response 3: As the groups were made up of children and young people with ages ranging from 6 to 20 years old, it was necessary to use instruments adapted according to their ages. In addition, as we were evaluating the cognitive (perception of aggression, perception of social support, self-esteem and self-concept) and neuropsychological (cognitive and emotional functions) domains, it was essential to have instruments, taking into account the directives of the respective articles, that could respond to all these variables.
- Procedure - what was the rationale for the split between the universities.
Response 4: We're not sure we understand the question. There was no split between the universities. The ISW is a Department of Lusófona University.
- P14 Discussion, 1st paragraph - statement re danger doesn't support the insight academics have into the behaviour of children within residential care. For instance - see the work of James Anglin Pain Based Behaviours.
Response 5: Thank you for your direction. We were not aware of the article. We have inserted the following paragraph (lines 455-456): “According to Anglin in 2003, the purpose of a RC is to create an artificial living environment that offers the youth residents an opportunity to develop a sense of normality.”
And we introduce the bibliographical reference:
Anglin, J.P. Understanding processes of residential group care for children and youth: Constructing a theoretical framework, International Journal of Child & Family Welfare 2003, 6(4), 141-150.
Comment 1. References have been adapted according to Healthcare standards.
We hope we’ve answered your questions. Once again, we thank you very much for the opportunity to improve our paper.

Reviewer 2 Report
Comments and Suggestions for Authors
I find this article to be fairly well thought through and there are aspects of it that I like. However, and this is, up to a point, a personal perspective, the article, as with many others on residential child care, betrays a very positivist, psychological (and primarily cognitive) orientation and as such, fails to recognise other more everyday literatures around residential care. To be fair, it does acknowledge this towards the end and recognises the need for more qualitative approaches, which take into account the views of children and staff who work in care settings.
I don’t think the article is helped by framing it in terms of the Unicef Report. Interestingly, I assumed it was referring to another Unicef report from 2024
https://www.unicef.org/eca/reports/keeping-families-together-europe
which manages to give very different figures regarding the use of residential care on a country basis. One of the things I found interesting about the report was that it was not Central or Eastern Europe that relied most on residential care but countries like Germany, Austria and the Netherlands, which have well-functioning residential care resources. It seems that Unicef tailor figures to support particular country narratives. I think it needs to be realised that Unicef promotes a particular ideological line, that is supported only by other ideological sources rather than other wider evidence or debate around the necessity in many cases of residential care..
It may be helpful to consider, for instance, Bruce Henderson’s recent book https://www.routledge.com/Challenging-the-Conventional-Wisdom-about-Residential-Care-for-Children-and-Youth-A-Good-Place-to-Grow/BHenderson/p/book/9781032564739?srsltid=AfmBOoqziQIM9iYz9PyZrqfRABJ3UurX2IznwBSQr5iU4pvkp5uK2P_y
On a more positive note, I like how the article acknowledges the shortcomings of treatment or ostensibly therapeutic models of practice and looks to one that is, to a large extent, activity based and rooted in an idea of salutogenesis and taking a preventative approach. This, I think, is a good starting point.
I liked that judo was part of the programme because martial and other physical arts can be prey to simplistic assumptions of them promoting violence. I agree that it can promote a positive body awareness and discipline. The arts too, are a useful component of any residential care facility’s overall programme.
While I agree that perhaps no other facility has combined exactly the same configuration of activities, others have done and do similar things.
I guess my cavil with the way that the findings of the programme are presented is that any improvements in functioning claimed can be attributed to the programmes (or its components), whereas, one of the major benefits of activities is that they allow relationships to form between children and adults that are not hierarchical and can then be of benefit to any improvements in functioning (and indeed perception). See https://cyc-net.org/cyc-online/cycol-0107-phelan.html
I note the 96 percent male sample size. I see that the authors mention this as a limitation but try and set it against a majority of males in residential care. But even so, the disparity is of a different magnitude and is (or may be) important to the findings
I wonder too about age. I note that age was taken into account in the assessment stage but I’m not sure if/how the activities varied according to age. I would imagine that there would be likely age differences in response. Perhaps, if so, these could be discussed.
The research seems to be well-enough conceived in the sense that it did have a baseline and sought to measure progress/impact over a period of three years
However, trying to identify outcomes from residential care studies is notoriously difficult as a whole number of variables contrive to influence what goes on in a centre. There can be a temptation to confuse correlation with causation. I am reminded of the Hawthorne effect on how attention to particular factors can lead to their enhancement. I have no trouble imagining that the kind of activities might have a beneficial effect on youngsters but I wonder if any activities might prove equally positive. How can it be said that this particular configuration of activities led to the improvements claimed? Moreover, the use of a control group presumes that those in the control groups did not have an intervention – but the very fact of residential care might be considered to be an intervention. Indeed, seminal literature on residential care ‘The Other 23 Hours’ suggests that what happens in everyday living is of at least as much import as any ‘programmatic’ intervention. The idea of life-space should be introduced.
Presumably, the interventions themselves were effected by care staff/social educators … what role did they play in the programme’s success or otherwise?
I have some difficulty with attempts to apply a scientific method to children’s care. Generally, I don’t think it takes us very far. In this case, for instance, I am intrigued by the fact that Verbal fluency scores seem to be better for control group. I wonder how that might be … perhaps we can only play hunches rather than try and come up with more ‘scientific’ answers.
I think the strength of the paper is that it sets out from a positive and thought-through starting point and the components of the programme make sense to me. I think the idea of a variety of ways of engaging with children is important but how this looks on a case-by-case basis can vary. And regardless of any ‘programmatic intervention, the important part of residential child care will be the kind of adult child (and child child) relationships that form there and how these are played out.
I think the paper is sufficiently well executed to merit publication. However, I think it could be improved if it were to engage with some of the literature on child and youth care or social education.
Author Response
We’d like to start by thanking you for your comments. They were extremely important for improving our paper.
Below, we'll respond to the comments you sent us.
- I don’t think the article is helped by framing it in terms of the Unicef Report. Interestingly, I assumed it was referring to another Unicef report from 2024
https://www.unicef.org/eca/reports/keeping-families-together-europe
which manages to give very different figures regarding the use of residential care on a country basis. One of the things I found interesting about the report was that it was not Central or Eastern Europe that relied most on residential care but countries like Germany, Austria and the Netherlands, which have well-functioning residential care resources. It seems that Unicef tailor figures to support particular country narratives. I think it needs to be realised that Unicef promotes a particular ideological line, that is supported only by other ideological sources rather than other wider evidence or debate around the necessity in many cases of residential care.
Response 1: Thank you for your comment, with which we fully agree, and for providing the link.
- It may be helpful to consider, for instance, Bruce Henderson’s recent book https://www.routledge.com/Challenging-the-Conventional-Wisdom-about-Residential-Care-for-Children-and-Youth-A-Good-Place-to-Grow/BHenderson/p/book/9781032564739?srsltid=AfmBOoqziQIM9iYz9PyZrqfRABJ3UurX2IznwBSQr5iU4pvkp5uK2P_y
Response 2: Thank you for your suggestion. We have inserted the following paragraph (Lines 74-76): “Also, according to Henderson (2023), several authors state that residential care as a treatment of choice with certain groups of needy children and youth, not a last resort intervention. Therefore, it is…”
And we introduce the bibliographical reference:
Henderson, B.B. Challenging the conventional wisdom about residential care for children and youth: A good place to grow. Routledge Advances in Social Work, London, 2023, UK. https://doi.org/10.4324/9781003435709
- On a more positive note, I like how the article acknowledges the shortcomings of treatment or ostensibly therapeutic models of practice and looks to one that is, to a large extent, activity based and rooted in an idea of salutogenesis and taking a preventative approach. This, I think, is a good starting point.
Response 3: Thank you for sharing.
- I liked that judo was part of the programme because martial and other physical arts can be prey to simplistic assumptions of them promoting violence. I agree that it can promote a positive body awareness and discipline. The arts too, are a useful component of any residential care facility’s overall programme.
Response 4: Thank you for your kind comment.
- While I agree that perhaps no other facility has combined exactly the same configuration of activities, others have done and do similar things.
Response 5: Thank you for your comment. We believe so, but unfortunately, we don't know of any establishment that has promoted this combination of activities, nor any specific study on these activities other than those presented. For Judo (Hokino and Casal, 2001; Jorge, 2022; Trusz et al., 2018); Arts (Freedman, 2003; Hetland et al., 2007; Ros, 2022; Sousa, 2003) and VR (Mikadze, 2014; Pontes and Hubner, 2008; Prigatano, 2005).
- I guess my cavil with the way that the findings of the programme are presented is that any improvements in functioning claimed can be attributed to the programmes (or its components), whereas, one of the major benefits of activities is that they allow relationships to form between children and adults that are not hierarchical and can then be of benefit to any improvements in functioning (and indeed perception). See https://cyc-net.org/cyc-online/cycol-0107-phelan.html
Response 6: We totally agree with your comment. To improve understanding of the work, in the lines 464 and 466, we insert the following information “Based on the analysis of the results obtained, even knowing that there are other variables that may have influenced the results, we can argue that the objective of D'AR-TE has been achieved.”
- I note the 96 percent male sample size. I see that the authors mention this as a limitation but try and set it against a majority of males in residential care. But even so, the disparity is of a different magnitude and is (or may be) important to the findings
Response 7: Thank you for your comment, which we agree with. We believe we have safeguarded this limitation.
- I wonder too about age. I note that age was taken into account in the assessment stage but I’m not sure if/how the activities varied according to age. I would imagine that there would be likely age differences in response. Perhaps, if so, these could be discussed.
Response 8: In fact, the ages were taken into account in the evaluations (initial and final) by adapting the assessment tests and in the activities: divided into two groups, younger (6-10 years) and older (11-20 years), with different times for judo (2 hours and 1 hour, respectively) and activities adapted according to age in the arts workshop and virtual reality (same scenario, but with different levels of difficulty in the tasks performed virtually). However, the objectives were similar (instruments and activities), only they were adapted, according to the literature, according to age.
- The research seems to be well-enough conceived in the sense that it did have a baseline and sought to measure progress/impact over a period of three years
Response 9: Thank you for your kind comment.
- However, trying to identify outcomes from residential care studies is notoriously difficult as a whole number of variables contrive to influence what goes on in a centre. There can be a temptation to confuse correlation with causation. I am reminded of the Hawthorne effect on how attention to particular factors can lead to their enhancement. I have no trouble imagining that the kind of activities might have a beneficial effect on youngsters but I wonder if any activities might prove equally positive. How can it be said that this particular configuration of activities led to the improvements claimed? Moreover, the use of a control group presumes that those in the control groups did not have an intervention – but the very fact of residential care might be considered to be an intervention. Indeed, seminal literature on residential care ‘The Other 23 Hours’ suggests that what happens in everyday living is of at least as much import as any ‘programmatic’ intervention. The idea of life-space should be introduced.
Response 10: We totally agree with this reflection. Because we believe that living space alone is a determining factor, we chose to have a control group as similar as possible to the experimental group and RC homes with similar operating dynamics. To make up the control group, we carried out a survey of shelters (similar operating dynamics) with a population as similar as possible to that of the experimental group (mostly boys, average age, reasons for signaling danger). We've put a paragraph (lines 144-148) in the article with this information: “The experimental group was constituted by children and young people with a promotion and protection measure who were in RC at the institution where D'AR-TE was implemented. To constitute the control group, a survey was carried out of RC homes with a population as similar as possible to that of the experimental group (mostly males, average of age, reasons for signaling danger).”.
- Presumably, the interventions themselves were effected by care staff/social educators … what role did they play in the programme’s success or otherwise?
Response 11: Thank you for your comment. Indeed, some professionals (judo master, music teacher, drama teacher, body expression teacher) were integrated into the course of D'AR-TE, but RC technicians and educators also took part in all the activities. To boost the involvement of these professionals with the children and young people, before the formal start of the project's activities, they visited the house and took part in various dynamics of RC's daily life.
- I have some difficulty with attempts to apply a scientific method to children’s care. Generally, I don’t think it takes us very far. In this case, for instance, I am intrigued by the fact that Verbal fluency scores seem to be better for control group. I wonder how that might be … perhaps we can only play hunches rather than try and come up with more ‘scientific’ answers.
Response 12: We understand your point perfectly. However, the only conclusion we've been able to reach is that this dynamic set of activities doesn't seem to enhance some of the skills (such as verbal fluency) and that there were other dynamics used by the control group that may have enhanced this skill.
- I think the strength of the paper is that it sets out from a positive and thought-through starting point and the components of the programme make sense to me. I think the idea of a variety of ways of engaging with children is important but how this looks on a case-by-case basis can vary. And regardless of any ‘programmatic intervention, the important part of residential child care will be the kind of adult child (and child child) relationships that form there and how these are played out.
Response 13: We totally agree and believe that these points are crucial to the success of residential care. The D'AR-TE project did not intend to override this unquestionable reality, but only to reinforce it through organized group dynamics (adult-child, child-child, young-young and adult-young).
- I think the paper is sufficiently well executed to merit publication. However, I think it could be improved if it were to engage with some of the literature on child and youth care or social education.
Response 14: New references have been introduced.
Comment 1. References have been adapted according to Healthcare standards.
We hope we've answered your questions. Once again, we thank you very much for the opportunity to improve our paper.

Reviewer 3 Report
Comments and Suggestions for Authors
The study addresses a relevant and novel topic through the D'AR-TE intervention model for young people in residential care, though it does present certain limitations, outlined below.
The project is well-explained, and specific examples of activities are included. However, it would be beneficial to delve deeper into how these activities are integrated into the participants’ daily lives. To facilitate the extrapolation of the model to other contexts, it would be useful to specify the location and conditions under which the activities were conducted (e.g., whether they were held in sports or community clubs, if they were specifically funded, or if they were exclusive to the project).
It would also be advisable to detail the sample selection process and how participants who left the project before completion were accounted for. Additionally, gathering perspectives from the participants and their surroundings could offer valuable insights into the project’s impact from the viewpoint of those directly involved. Furthermore, it is important to explain the gender disparity in the sample and consider any cultural and contextual factors that could influence the results.
Although the study mentions the limitation of the sample size, the results are presented as if they came from a larger, more diverse group, potentially generalizable to other contexts. It would be helpful to clarify that the results are specific to this particular project and context, noting that extrapolating these findings to other settings should be done with caution.
Author Response
We'd like to start by thanking you for your comments. They were extremely important for improving our paper.
Below, we'll respond to the comments you sent us.
- The project is well-explained, and specific examples of activities are included. However, it would be beneficial to delve deeper into how these activities are integrated into the participants’ daily lives. To facilitate the extrapolation of the model to other contexts, it would be useful to specify the location and conditions under which the activities were conducted (e.g., whether they were held in sports or community clubs, if they were specifically funded, or if they were exclusive to the project).
Response 1: Thank you for the directives. Lines 189-190- have been introduced -“The implementation of the D'AR-TE project, which was part of the children's and young people's daily routines and took place on RC premises, is shown in Table 3.”
- It would also be advisable to detail the sample selection process and how participants who left the project before completion were accounted for. Additionally, gathering perspectives from the participants and their surroundings could offer valuable insights into the project’s impact from the viewpoint of those directly involved. Furthermore, it is important to explain the gender disparity in the sample and consider any cultural and contextual factors that could influence the results.
Response 2: Thank you for your questions and advice.
We will answer this question in points:
2.1. To answer the first question, the following paragraph has been introduced (lines 144-148): “The experimental group was constituted by children and young people with a promotion and protection measure who were in RC at the institution where D'AR-TE was implemented. To constitute the control group, a survey was carried out of RC homes with a population as similar as possible to that of the experimental group (mostly males, average of age, reasons for signaling danger).”
2.2. Participants who left the project before its completion were not included in the data presented in the study.
2.3. Collecting the perspectives of the participants and their environment effectively provided valuable information on the impact of the project from the point of view of the people directly involved. These results have been interpreted in other works, such as a master's thesis presented at the Lusófona University by Mara Almeida Ricardo “Promoção do suporte social e dinâmica relacional de crianças e jovens em contexto de Acolhimento Residencial: o impacto do Projeto D´AR-TE”/“Promoting social support and relational dynamics of children and young people in residential care: the impact of the D'AR-TE Project”. However, in this article we have chosen not to make it more complex, but we fully understand the question you raise, and it would be an asset.
2.4. In Portugal, most RC’s are not mixed. This directive has already been published, but it is still very recent. Therefore, the gender limitation was difficult to overcome. We tried to explain in the lines 485-488.
- Although the study mentions the limitation of the sample size, the results are presented as if they came from a larger, more diverse group, potentially generalizable to other contexts. It would be helpful to clarify that the results are specific to this particular project and context, noting that extrapolating these findings to other settings should be done with caution.
Response 3: Thank you for your comment. To try to record this difficulty, we have included lines 478-481 “The new RC intervention model proved to be effective in these children and young people, however, the extrapolating these findings to other settings should be done with caution. The follow-up evaluation (one year after the end of the project) is still running.”
Comment 1. References have been adapted according to Healthcare standards.
We hope we've answered your questions. Once again, we thank you very much for the opportunity to improve our paper.

Reviewer 4 Report
Comments and Suggestions for Authors
The article presented is relevant to the field of social intervention in residential care. However, it could be improved in several aspects:
The theoretical framework does not include enough references to adequately support the study's claims and theoretical foundations. The article should expand the theoretical framework by integrating more relevant and recent studies that better substantiate the claims and the basis of the intervention. Although the article is well-written, some paragraphs could be revised to make the reading more accessible to a broader audience.
While the results are clear, the discussion could benefit from greater depth in the critical analysis of the findings, including a more exhaustive comparison with previous studies and a more detailed reflection on the practical and theoretical implications of the findings.
It would be helpful to include a broader discussion on the long-term impact of the D'AR-TE project.
The sample size is limited, which could affect the generalizability of the results to other populations of children and young people in residential care settings. The study does not mention a long-term follow-up after the intervention, which is important to assess the sustainability of the observed changes.
Although a control group is included, it would be interesting to expand the control criteria to better understand how different variables (such as the intensity of the intervention or the psychological profile of the participants) affect the results. The theoretical framework does not include enough references to adequately support the study's claims and theoretical foundations.
Author Response
We'd like to start by thanking you for your comments. They were extremely important for improving our paper.
Below, we'll respond to the comments you sent us.
- The theoretical framework does not include enough references to adequately support the study's claims and theoretical foundations. The article should expand the theoretical framework by integrating more relevant and recent studies that better substantiate the claims and the basis of the intervention. Although the article is well-written, some paragraphs could be revised to make the reading more accessible to a broader audience.
Response 1: Thank you for your directive. To improve the theoretical underpinning of the article, we have inserted the following information: Lines 65-76- “In 2022, Cameron-Mathiassen's team conducted a systematic literature review with the aim of identifying the experience of living in CR. They searched five relevant databases for qualitative empirical studies published between 1990 and January 2020. The results were very heterogeneous. For some, RC is referred to as an experience that promotes care, stability, security and a perception of support, while others report experiences of not being listened to or understood by the institution and a poor perception of well-being. Relationships with peers were also experienced both positively, with the development of friendly and supportive relationships, and negatively, with the peer group accepting bullying and violence as normative behaviors. Also, according to Henderson (2023), several authors state that residential care as a treatment of choice with certain groups of needy children and youth, not a last resort intervention. Therefore, it is…”
The two bibliographical references have been introduced:
Cameron-Mathiassen, J.; Leiper, J.; Simpson, J.; McDermott, E. What was care like for me? A systematic review of the experiences of young people living in residential care. Children and Youth Services Review 2022, 138, 106524. https://doi.org/10.1016/j.childyouth.2022.106524
Henderson, B.B. Challenging the conventional wisdom about residential care for children and youth: A good place to grow. Routledge Advances in Social Work, London, 2023, UK. https://doi.org/10.4324/9781003435709
- While the results are clear, the discussion could benefit from greater depth in the critical analysis of the findings, including a more exhaustive comparison with previous studies and a more detailed reflection on the practical and theoretical implications of the findings.
Response 2: Thank you for your comment. However, as this is a pilot project, it is difficult to make a comparison, and we have chosen to discuss the data between the evaluation periods (initial and final) and groups (experimental and control). As for the practical implications of the results, we have included the following lines 478-481- “The new RC intervention model proved to be effective in these children and young people, however, the extrapolating these findings to other settings should be done with caution. The follow-up evaluation (one year after the end of the project) is still running.”
- It would be helpful to include a broader discussion on the long-term impact of the D'AR-TE project.
Response 3: We agree, but now we are still following up the project (1 year). As soon as we have the results, we hope to be able to share them with you.
- The sample size is limited, which could affect the generalizability of the results to other populations of children and young people in residential care settings. The study does not mention a long-term follow-up after the intervention, which is important to assess the sustainability of the observed changes.
Response 4: Thank you for your comment, which allowed us to include lines 480-481 “The follow-up evaluation (one year after the end of the project) is still running.”
- Although a control group is included, it would be interesting to expand the control criteria to better understand how different variables (such as the intensity of the intervention or the psychological profile of the participants) affect the results. The theoretical framework does not include enough references to adequately support the study's claims and theoretical foundations.
Response 5: Thank you for raising such important questions that we will certainly take into consideration in a future study. We have included some references, such as:
Cameron-Mathiassen, J.; Leiper, J.; Simpson, J.; McDermott, E. What was care like for me? A systematic review of the experiences of young people living in residential care. Children and Youth Services Review 2022, 138, 106524. https://doi.org/10.1016/j.childyouth.2022.106524
Henderson, B.B. Challenging the conventional wisdom about residential care for children and youth: A good place to grow. Routledge Advances in Social Work, London, 2023, UK. https://doi.org/10.4324/9781003435709
Comment 1. References have been adapted according to Healthcare standards.
We hope we've answered your questions. Once again, we thank you very much for the opportunity to improve our work.

Round 2
Reviewer 3 Report
Comments and Suggestions for Authors
The authors included all the coments that I suggest in the last revision